# “Seeing Is Believing”: How Neutron Crystallography Informs Enzyme Mechanisms by Visualizing Unique Water Species

**DOI:** 10.3390/biology13110850

**Published:** 2024-10-22

**Authors:** Qun Wan, Brad C. Bennett

**Affiliations:** 1Jiangsu Provincial Key Lab for Solid Organic Waste Utilization, Key Lab of Organic-Based Fertilizers of China, Jiangsu Collaborative Innovation Center for Solid Organic Wastes, Educational Ministry Engineering Center of Resource-Saving Fertilizers, Nanjing Agricultural University, Nanjing 210095, China; 2Department of Biological and Environmental Sciences, Samford University, Birmingham, AL 35229, USA

**Keywords:** neutron diffraction, deuterium, hydronium, protein structure, enzyme catalysis, H/D exchange

## Abstract

Neutron crystallography is a powerful technique for protein structure determination that excels in locating the “lighter” atoms, such as hydrogen. Unlike X-ray crystallography, which scatters primarily from heavier atoms like carbon, neutron scattering provides direct information about the positions of hydrogens, and thus can inform on water molecules vital for enzyme function. By exchanging deuterium for hydrogen, one can significantly enhance the signal in neutron experiments. With neutrons, we can precisely map the locations of these “heavy” water molecules (D_2_O), which reveal their role in hydrogen bonding networks and substrate interactions. This aids in deciphering enzyme mechanisms, enhancing drug design, and informing biological function, making neutron crystallography an invaluable tool in structural biology. In the last 15 years, several neutron structures of enzymes have identified unique water species beyond D_2_O molecules, such as hydronium (H_3_O^+^ or D_3_O^+^) and even sole deuterons (D^+^). These are thought to be more transient and dynamic in protein structures, thus they could inform more directly on mechanisms than standard water molecules. This review summarizes four of these studies and provides a reflection on our mentor, Dr. Chris G. Dealwis, who recently passed away but who was an important figure in neutron crystallographic studies of enzymes.

## 1. Introduction

In terms of efficiency in macromolecular structure determination, you can do better than neutron crystallography (NC). For experimental solutions, it is certainly eclipsed by the “big three” methods in structural biology: X-ray crystallography (XRC), electron cryomicroscopy (cryoEM), and nuclear magnetic resonance (NMR) spectroscopy. However, neutrons are unique probes of macromolecular structure in that they allow direct observation of hydrogen (H) atoms. Although it is the lightest atom, H is also the most abundant in macromolecules, composing ~50% of the atomic content in a protein. Far from being inert structural components in macromolecules, the position, movement, sharing, and exchange of H atoms are essential to most biochemical processes including protein:protein interactions and enzyme catalysis. As X-rays are scattered by the electron cloud of atoms, and scattering power increases with the atomic number (*Z*), they are only scattered very weakly by H’s lone electron. Only electron density maps at ultrahigh resolutions (~1.0 Å or better) can provide information on some of the H positions. This resolution is remarkably difficult to achieve, as only ~0.5% of structures deposited in the Protein Data Bank (PDB; RCSB.org [1]) are in this range. Fortunately, neutrons are scattered by atomic nuclei, and the strength and coherence of their scattering profile is directly related to the content and spin state of the nuclei. Scattering power is independent of the atomic number, with scattering lengths and coherence dramatically changing as one scans across the periodic table (Table 1 [2]). Historically, very large crystal volumes (>1 mm^3^, or at least 10× the volume of crystals suitable for XRC) and long data collection times (weeks to months) were required to achieve data completeness and provide a strong enough scattering intensity (signal) to be measured by neutron detectors [3]. By exchanging H for deuterium (^2^H, or D), the signal-to-noise ratio in an NC experiment can be tremendously enhanced, as D scatters with the strength of heavier elements such as carbon. H/D exchange for NC is typically achieved by either growing crystals from or, more frequently, soaking crystals in deuterated buffer components and mother liquor. This will result in H/D exchange at some of the chemically exchangeable positions, such as on side-chain hydroxyl and backbone amide groups. Alternately, one can use perdeuteration, where D is incorporated into the protein while it is being synthesized. Although more expensive and often labor intensive, it guarantees H/D exchange at all chemically *non-exchangeable* positions, such as on side-chain methyl groups. Regardless of the process by which D is incorporated into the protein, care should be taken to maintain the crystal in a deuterated environment prior to the NC experiment as this will prevent back-exchange to H at chemically exchangeable positions. (Note: importantly, for perdeuterated samples, deuterons incorporated at non-exchangeable positions will not back-exchange.) Coupling H/D exchange with NC is the basis by which meaningful observations can be made regarding protonation states, hydrogen bonding patterns, and solvent structure in macromolecules.

By the end of 2008, there were only 28 macromolecular structures deposited in the PDB that had been determined by NC. The first report of a neutron structure for myoglobin was published in 1969, and there was a very slow drip of new neutron structures over the next 4 decades. We can term this initial period in the history of NC as the “discovery by proof-of-principle” phase, where (unsurprisingly) most of the models deposited and published were the 1st neutron structures of a particular macromolecule. Around 2009–2010, a major inflection event occurred and nearly 200 new structures have been added to the PDB since then [3,4]. As of July 2024, there are 226 total NC structures, with 208 of those being for proteins (RCSB.org; [1]). One-quarter of all NC structures have been deposited since 2020. Much of this explosive growth is due to the consistent availability of seven neutron diffractometers worldwide dedicated to structural biology: BIODIFF (Germany), BIX-3 and -4 (Japan), iBIX (Japan), IMAGINE (USA), LADI-III (France), and MaNDi (USA). New diffraction instrumentation is being developed at each of the facilities where these stations reside with more powerful particle beams and more sensitive detectors, allowing complete data sets to be collected from smaller volume crystals in days (for an example of the immense technical improvements at NC beamlines, see [5] and their description of MaNDi). Of immense interest is the European Spallation Source (ESS), recently constructed in Lund, Sweden, and expected to have first beam on target by 2025, with initial NC experiments the following year (https://europeanspallationsource.se/ accessed on 1 July 2024). We term this current time period (2009–present) as the “discovery by comparison” phase, where most (but not all) of the structures that are being deposited and published are for macromolecules that have a previously deposited neutron model or where multiple models are being deposited within one study. Additionally, much like the field of electron cryomicroscopy (cryoEM), we are also experiencing a “resolution revolution” in NC structures; many of the deposited structures in this current phase are sub-2Å resolution. A recent neutron structure of High Potential Iron-Sulfur Protein (HiPIP) approach 1Å resolution, allowing direct observation of highly specific and precise phenomena, such as deviations from peptide bond planarity [6].

Fortunately, even at modest resolutions (~2.5 Å), macromolecular structures determined by NC can readily allow one to distinguish between H and D in nuclear density maps, as H appear in negatively contoured m*F_o_*–D*F_c_* maps whereas D atoms stand out as strong peaks in positively contoured m*F_o_*–D*F_c_* maps [7]. Contouring between positive and negative levels in omit maps are a particularly useful way for modeling H and D at exchangeable positions. A commonplace strategy now is to collect high-resolution XRC data on an isomorphous crystal, preferably one from the same crystallization drop, so one can use a joint X-ray/neutron structure refinement deployed in program suites such as Phenix [8]. The electron density maps are used to model and confirm the position of the “heavier” atoms in macromolecules (C, N, O, S, P, and most metals), whereas the nuclear density maps can reliably allow the modeling of H/D. Most non-hydrogen atoms found in protein structures are strong, coherent scatterers of neutrons and theoretically should be observable and able to be modeled using nuclear density data. However, if covalently bonded to hydrogen, the massive incoherent scattering cross section for neutrons by hydrogen may obscure and reduce the nuclear density peaks of the non-hydrogen atoms to which they are bonded. Thus, subsequent refinement of the model is critical for establishing the occupancy of H vs. D at a specific position. Of course, this is necessary to determine the actual level of H/D exchange that has occurred. For proteins, the backbone amide exchange rate can be directly observed from nuclear density maps. This is a measure of exchange propensity, which itself can inform on the rigidity or flexibility of the protein [9]. An extension of analyzing nuclear density maps for H/D exchange is that they can also reveal side-chain protonation states for ionizable residues such as Asp and His and protonation states of bound ligands (Figure 1). Bonding interactions can be mapped completely with this layer of information and, importantly for enzymes that utilize acid-base catalysis, proton transfer pathways can be gleaned. Lastly, water molecules can be completely modeled as three atom species, revealing orientation of the molecule itself and intricate hydrogen bonding patterns.

Related to this, in a handful of protein structures determined by NC, “water-adjacent” ion species have been modeled that may play a role in stability and/or catalysis. These include hydronium (D_3_O^+^), deuteroxide (OD^−^), and deuterons (D^+^). In this review, we will specifically focus on four enzymes, where one or more of these species has been modeled into NC structures and demonstrate how these species may be of functional significance. Where appropriate, we will briefly address some practical considerations, limitations to interpretation, and potential complementary techniques for verification of these species.

This is not meant to be an exhaustive review of NC as a technique or an all-encompassing survey of the immense number of important macromolecular structures probed by NC. For a broader perspective and appreciation of the use of NC in structural biology, the reader is encouraged to peruse one of several excellent reviews that have been published in the last 10 years. These provide information not found in the present work, such as but not limited to: a history of NC; a detailed description of NC instrumentation, facilities and software; differences amongst neutron diffraction methods (such as quasi-Laue and time-of-flight); macromolecular crystallization for neutron diffraction; H/D exchange and perdeuteration; lists of neutron structures and their associated statistics; and a compendium of case studies that exemplify the power of NC to determine molecular structures and answer specific biochemical questions [3,4,10,11,12,13,14,15].

## 2. Identification of Water (H_2_O) Molecules and Hydronium (D_3_O^+^), Deuteroxide (OD^−^), and Deuteron (D^+^) Ions in Enzyme Nuclear Density Maps

In electron density maps produced by X-ray crystallography, well-resolved water molecules as well as hydroxide or hydronium ions will all appear as singular, spherical peaks and are essentially indistinguishable from each other. This peak is of course representative of the oxygen atom, as it scatters X-rays strongly due to its high electron content relative to H atoms. Indeed, H atoms and protons (H^+^) are rarely directly observed in electron density maps; typically, only if the resolution is sub-atomic (<1 Å) and if the H atom is covalently bonded to a heavier atom in a rigid part of the structure. Even then, H bonded to polar atoms (namely, N and O) may still resist visualization. Especially if H/D exchange has occurred, neutrons are ideal for identifying and distinguishing between water (as D_2_O), hydronium (as D_3_O^+^), hydroxide (as deuteroxide, or OD^−^), and protons (as a deuteron, or D^+^) in protein structures. (Note: We will use the terms D_2_O, deuteroxide, and deuteron to describe the deuterated forms of these species as identified in nuclear density maps. However, we will use the term hydronium to refer to the deuterated water ion, D_3_O^+^. Though counterintuitive as it technically describes H_3_O^+^, it is more frequently used than the rather strange term, deuteronium.) The neutron scattering strengths for D and O are nearly identical (Table 1). A well-resolved deuterated water molecule (D_2_O) in a nuclear density map will appear as a larger, continuous spherical peak of density compared to the same water molecule in an electron density map. Some especially well-ordered D_2_O molecules will have three-lobed nuclear density peaks with a bend that are reminiscent of a crescent or boomerang. This is beautifully demonstrated in the 1.1 Å-ultrahigh-resolution nuclear density maps of the small hydrophobic protein, crambin [16]. For hydronium (D_3_O^+^) nuclear density peaks, their profile also is three-lobed, but the overall shape is pyramidal, with the three D atoms constituting the lobes and emanating out from the central oxygen. Clear examples of this are found in the 1.3–1.4 Å neutron structures of redox forms of the small iron-sulfur cluster protein, rubredoxin [17]. Several criteria are used to verify the identity of the water species in nuclear density maps, so as to distinguish between D_2_O, D_3_O^+^, OD^−^, and D^+^: (1) as mentioned above, the shape of the nuclear density, (2) the bonding and/or coordination contacts and binding distances in the binding site, (3) occupancy refinement of the atoms in the species, (4) examination of the m*F_o_*–D*F_c_* maps post-modeling and refinement to ensure difference density has been accounted for and no negative density has emerged, and (5) specifically for verification of a D^+^ would be the absence of a difference peak in electron density maps (as a deuteron would only very weakly scatter X-rays and are not observable in most electron density maps). Hydroniums and other unusual water-related species have been identified in a growing number of protein structures determined by NC. Neutron structures of four specific enzymes will now be highlighted.

### 2.1. Xylose Isomerase

The first direct observations of a hydronium, deuteroxide, and deuteron in a protein structure were all made from NC analysis of xylose isomerase (XI) from *Streptomyces rubiginosus* [18,19,20]. XI catalyzes the conversion of aldose sugars (such as glucose) to ketose sugars (such as fructose), and its mechanism goes through three major steps: aldose ring opening, aldehyde-to-ketone isomerization, and ketose ring closure [21]. There is biotechnological interest in XI as it has proven useful in production of biofuels from genetically engineered yeast [22] and in sugar conversion to make commercially important sweeteners, amongst many other applications [23]. XI is assisted in catalysis by two metals (typically Mg^2+^) that bind in specific sites (termed M1 and M2) within its active site; these are coordinated by several acidic residues (Asp and Glu). In addition, a catalytic water is positioned near the M2 site and forms a coordination bond with it. Prior to crystallization, these intrinsically bound metals are removed by chelation so that different metals and specific sugar substrates/intermediates/products can be soaked into apo crystals. The four initial NC structures of XI with various combinations of metals and sugars bound revealed protonation states of the ionizable residues in the active site (namely, His 54 and Lys 289) as well as the order and direction of proton (as D atoms) movements in the three-phase mechanism [18,19]. One of the structures is with Mg^2+^ and fructose product bound in the active site, which represents the mechanistic step after isomerization but prior to re-formation of the ring and dissociation of the fructose product. Instead of the catalytic water as observed as a D_2_O molecule in the pre-isomerization structures, a deuteroxide (OD^−^) was observed coordinated to the Mg^2+^ in the M2 site (Figure 2C). Although this was the first direct observation of a deuteroxide species in a protein structure, it uncovered a new aspect of the mechanism. A deuteroxide has also been identified in a 1.5 Å-resolution neutron structure a copper-containing nitrite reductase. Like XI, the deuteroxide forms a coordination bond with a metal (Cu^2+^ instead of Mg^2+^). Taken together with all the structures for XI, this deuteroxide coordination suggested that the catalytic water donates the proton to the O1 atom on the sugar, triggering isomerization, and changes the water to a transient hydroxide ion (observed as a deuteroxide in the maps) [19,24].

In a follow-on study, intrinsic metals were removed from XI prior to crystallization but this time no extrinsic metals were soaked into the crystals prior to neutron diffraction [20]. Thus, these crystals should be “true” apo forms, with no metals or substrates bound. Crystals were grown in the same composition D_2_O-based buffers as before with one exception: two pH values were tested, 7.7 as before and 5.9. (Note: In D_2_O-based buffers, the pH should be reported as the pD, which is 0.4 pH units higher (more alkaline) than the actual pH. This is due to the slightly shorter length and thus stronger covalent bond made by D atoms compared to H. We use both terms in this review as some studies report pH and others report the pD; thus, we use whatever values the authors of the respective studies used. Please refer to the cited articles for clarity.) Using mass spectrometry (MS), it had been previously shown that the catalytic metal ions are expelled at acidic pH, and this is associated with a loss of enzymatic activity [25]. In the pH 7.7 apo crystals, a hydronium ion (D_3_O^+^) was observed binding in the M1 site (the M2 site was occupied by water) (Figure 2A). Assignment of the hydronium was supported by the trigonal pyramid shape of the nuclear density in an m*F_o_*–D*F_c_* omit map, subsequent occupancy refinement showing that all three D atoms have an occupancy of 1.0, and inductively coupled plasma MS (ICP-MS) that verified no metals were bound to the protein used for crystallization. Strikingly, in the pH 5.9 structure, the hydronium has been replaced (“dehydrated”) to a lone deuteron (D^+^) (Figure 2B). Its presence was confirmed by the shape and location of the density peak, occupancy refinement, and the absence of a peak at this location in the corresponding electron density maps. The active site has essentially collapsed around the deuteron, and it forms very short (potentially low barrier) hydrogen bonds with Asp and Glu residues in the active site. The overall shape of the active site has changed markedly to where it cannot accommodate a metal. Indeed, metal-soaking experiments were performed on crystals at pH 5.8, XRC data were collected, and only 60% occupancy could be accomplished and that only for the M2 site. No metals could displace the deuteron to bind at the M1 site. This provided a basis for the loss in catalytic activity at acidic pH. Furthermore, this provided the first observation of a hydronium and a deuteron in a protein. By using a “pH trapping” technique, the hydronium at pH 7.7 could be exchanged for the deuteron at pH 5.9. The pH 5.9 structure also served as the first observation of a deuteron involved in chelating oxygen ligands in a protein [20].

QM/MM calculations were performed independently to determine if the isolated hydronium in the active site of XI was stable [26]. These produced an energy profile for the three short hydrogen bonds formed between the hydronium and the acidic residues in the active site, revealing that the energy minimum was located at the hydronium and suggesting that the hydronium indeed was stable in the XI active site. The authors argue that the four acidic residues (Asp and Glu) in the active site stabilize the ionized hydronium (D_3_O^+^); in fact, it appears that a multitude of acidic residues as hydrogen bonding partners is a requisite for a hydronium to be able to be stable and long-lived *in a protein interior* [26]. This is contrasted with the several hydroniums observed in ultrahigh-resolution NC maps of rubredoxin; these are all on the protein surface and are in hydrogen bonding networks with surrounding water molecules [17].

### 2.2. Bilin Reductase PcyA

Phycocyanobilin:ferrodoxin oxidoreductase (PcyA) converts biliverdin IX*a* (BV) into phycocyanobilin (PCB), an important light-harvesting pigment found from plants to cyanobacteria [27]. These pigments contain a chromophore of four pyrrole rings, and PcyA catalyzes a two-step sequential reduction of vinyl groups. As this activity involves the transfer of two protons, there are at least two critical ionizable residues in the vicinity of the chromophore (His 88 and Asp 105), and a water molecule has been implicated to be essential for catalysis [28] but not unambiguously identified in previous XRC studies, an NC analysis of a cyanobacterial PcyA was undertaken [29]. Just below the area where the pyrrole chromophore resides and the site of catalysis, there appeared to be a water molecule that bridged His 74 and His 88, the aforementioned essential residue. In m*F_o_*–D*F_c_* omit maps near His 88, a strongly positive peak was observed near its Nε atom, initially interpreted to be a deuterium covalently bonded to the Nε and thus meaning it is protonated and, due to the Nδ atom also being protonated, the His is positively charged. However, the position of the peak was such that it was not an ideal bonding distance from the Nε atom or in plane with the His imidazole ring. The authors re-interpreted this peak as a deuterium directly bonded to an adjacent water molecule, thus making it a hydronium (a D_3_O^+^) (Figure 2D). When the water is also omitted, the m*F_o_*–D*F_c_* omit map has a flattened, triangular pyramidal shape and, upon modeling a hydronium into the density, no residual density remains. Upon refinement, the occupancies of all three deuterium atoms converged to 1.0. When the authors attempted to model partially occupied D_2_O molecules in this peak, significant residual density remained. Taken together, this suggests that a hydronium is what bridges the two His residues. Furthermore, they proposed a mechanism in which the His 88 is a proton donor to one of the BV rings and is concomitantly protonated itself by the hydronium. In contrast to the hydronium molecules identified in the rubredoxin [17] and XI [20] neutron structures, the hydronium modeled in PcyA is implicated to have an important functional role in catalysis [29]. Similar to the hydronium in XI, this is also found within the protein interior.

QM/MM calculations were performed to approximate the stability of the hydronium in PcyA [26]. The H_3_O^+^ in PcyA releases a proton to the His 88 Nε atom, thus revealing that the hydronium moiety is unstable. In fact, a potential energy profile of the hydrogen bonding between the hydronium and either the His 74 Nδ atom or the His 88 Nε atom showed energetic favorability for the proton involved to be located nearer (and presumably bonded) to the imidazole ring of the histidines. The energy profile is more favorable (a greater −ΔG) for the proton to transfer to His 88 than His 74. Altogether, the calculations suggest that, instead of a hydronium bridging two neutral His residues as modeled in the neutron structure, a water molecule forms hydrogen bonds between a charged His 88 and a neutral His 74. The authors note that the hydronium binding site in PcyA is devoid of acidic residues. With the p*K*_A_ of H_3_O^+^ being −1.7 and the p*K*_A_ of a His Nε atom being ~7, this absence of Asp or Glu residues means that significant p*K*_A_ shifts of either the hydronium or His 88 are essentially impossible; in other words, it would be unlikely for a water molecule to attract and bind an additional proton in this particular binding site. With the hydronium modeled in the XI structure, its binding site is rich with acidic residues (Asp 245 and 287, Glu 181 and 217) [20]. This likely increases the p*K*_A_ of the hydronium from -1.7 to above the p*K*_A_ for the Asp/Glu residues, their p*K*_A_ values typically being 3–4, and stabilizing the additional proton on the hydronium [26]. This isolated hydronium in XI may serve a functional structural role in binding and stabilizing the negative charges on the Asp and Glu acidic residues and maintaining the binding site for metals. The hydronium modeled in PcyA would not need to serve this purpose, thus this isolated species may not be stable. Perhaps a short-lived, transient hydronium species can and is necessary to act as a proton shuttle between His 74 and His 88; however, it seems unlikely that this species could be “trapped” and what is observed in the nuclear density maps. The crystallization was performed at p*D* 6.3, and the data collection took place over several days at room temperature.

Further NC structures of PcyA were performed for two site-directed mutants, I86V and D105N, that affect BV absorption spectra and catalysis [30]. Strong m*F_o_*–D*F_c_* omit nuclear density was observed between His 74 and His 88, suggesting a water molecule bonding with these residues as seen in the WT structure. Interestingly, it appears the water molecule rotates its orientation between the two mutants, resulting in a charged His 88 (and neutral His 74) for the I86D structure with a charged His 74 (and neutral His 88) for the D105N structure. Indeed, a water molecule (D_2_O) is modeled at this site for both the I86D and D105N structures and the authors argue that it is “fixed with stability”. This is contrast to the WT structure, where this water is either a short-lived hydronium or a water molecule with dual orientations and thus is not in a fixed position [30].

### 2.3. Glycoside Hydrolase: Xylanase (Xyn II)

Xyn II is a specific type of xylanase belonging to family 11 glycoside hydrolases. It is secreted from *Trichoderma reesei*, a well-known fungus for its cellulase and hemicellulase activities [31]. Xyn II breaks down the β-1,4-glycosidic bond in xylan, the backbone of hemicellulose, into xylose monomeric units. Its efficient hydrolytic activity makes it a valuable enzyme in various industrial applications, such as biofuel and paper production [32]. The neutron crystallographic structure of *T. reesei* Xyn 11 in an acidic environment (pD 4.8) shows that the catalytic acid, Glu 177, has a downward conformation, which is protonated and H-bonded with a nearby water molecule [33]. This water, modeled as a D_2_O in the maps, is proposed to be a hydronium (D_3_O^+^) and can serve as a proton donor to the Glu 177 (Figure 2E). Indeed, in m*F_o_*–D*F_c_* omit nuclear density maps at pD 4.8, a strongly positive peak is observed adjacent to the Glu 177 carboxylate, modeled as a proton, and a strong hydrogen bond is observed between the Glu 177 and the nearby water. In contrast, the neutron crystallographic structures at pD 6.2 and pD 8.9 show that Glu 177 has the upward conformation and is deprotonated (thus, negatively charged). These NC structures demonstrate that this general acid has two alternate conformations with different protonation states. Continuum electrostatics calculations estimate that this Glu has a significantly higher p*K*_A_ value in the downward conformation than the upward conformation. Thus, it is proposed that the carboxylate side chain of Glu 177 could be a general base to accept a proton from solvent (possibly from a D_3_O^+^) and rotate to its alternate conformation to donate this proton to the glycosidic oxygen atom as the acid. Such side-chain rotation only needs a free energy of 4.3 kcal/mol, which is in the range of molecular motion. It is important to note that the authors merely propose the existence of a transient D_3_O^+^ in this structure and do not model this water molecule as a hydronium; this is in contrast to the hydroniums identified and modeled in nuclear density for XI [20] and PcyA [29], as explained above. The NC structures and the MD calculations explain how this glycoside hydrolase is so efficient in its digestion of polysaccharides to monosaccharides [33].

Following the above studies, more NC structures at different pD environments reveal the important interaction between Glu 177 and the nearby Tyr 88 [34]. The hydroxyl H atom of the Tyr 88 side chain forms an H-bond interaction with the carbonyl oxygen atom of Glu 177 side chain. In the acidic environment, this H-bond is broken when Glu 177 is in the downward conformation. In the alkaline environment, the H-bond is stabilized with short distance and Glu 177 can hardly rotate to perform its function. In its optimal pH (pD 5.4), this H-bond is the weakest with long distance and Glu 177 can rotate quickly for catalysis. Constant pH MD simulations are consistent with the NC structures that Glu 177 rotation is regulated by pH with different protonation states. The above results demonstrate that the pH-dependent activity of glycoside hydrolase is due to the H-bond interactions among the catalytic acid and its nearby residue and solvent [34]. Modifying the H-bond interactions could help to change the pH optima for better industrial application.

### 2.4. Dihydrofolate Reductase

Dihydrofolate reductase (DHFR) catalyzes the NADPH-dependent reduction of dihydrofolate to tetrahydrofolate and is required for specific nucleotide and amino acid biosynthetic pathways. It is conserved across life and has served as a target for antimicrobial, antifungal, antiparasitic, and anticancer drugs for decades [35,36]. The enzyme-catalyzed mechanism involves a concomitant two-step reduction of the substrate DHF, with protonation of the N5 atom and hydride transfer from NADPH across the double bond of C6-C7. The enzyme progresses through a number of kinetic steps, with a rapid chemical step (also known as the hydride transfer rate), with the rate-limiting step being product release [37]. Data from several methodologies including molecular dynamics, kinetic analysis, NMR, and crystallographic studies have shown that protein motions and dynamics are intimately linked to and supportive of enzyme catalysis; this has been most extensively reported for the DHFR from *Escherichia coli* (*ec*DHFR) [38]. Protonation of the DHF N5 precedes hydride transfer from NADPH; however, there is no amino acid in the vicinity of N5 that can serve as a proton donor. The closest is Asp 27, which is ~6 Å away. Due to this distance and the persistence of water molecules nearby to the N5 in X-ray crystal structures and molecular simulations, focus shifted to solvent being the proton donor to the DHF substrate [39,40,41,42,43].

To provide direct evidence of a water molecule playing the role of proton donor in the DHFR mechanism, we solved several ultrahigh-resolution X-ray structures at different temperatures and two neutron structures at different pH values (7.0 and 4.5) for *ec*DHFR [44,45]. These structures all represented a “pseudo-Michaelis” complex, with the weak substrate folate and product cofactor NADP^+^ bound. A hallmark of the *ec*DHFR structure is the highly flexible Met20 loop (residues 9–24) that changes conformation as catalysis proceeds from apoenzyme (open) to the Michaelis complex (closed) and finally to a product-bound complex (occluded). The loop is adjacent to the active site where DHF and NADPH bind and acts as a dynamic lid. Thus, this conformational cycle is thought to be regulatory in nature, as the alteration in loop structure allows the enzyme to progress through catalysis: the loop is open when substrate and cofactor need to bind, the loop closes when the chemical steps occur (protonation and hydride transfer), and the loop occludes the active site to encourage product (THF) release so the cycle can start over [40]. In one of the ultrahigh-resolution X-ray structures (1.05 Å resolution at 277 K; PDB ID 4RGC), a partially occupied water molecule could be modeled within hydrogen bonding distance of the N5 atom of folate [44]. The basis for the partial occupancy of this water appears to be due to the conformational flexibility of the Met 20 side chain, which could be modeled as two alternate conformers rotating about the Sδ atom. The major conformer (~60% occupancy) places the terminal Cε methyl group of the side chain to where it sterically restricts a water molecule near the N5 of the DHF substrate. The minor conformer (~40% occupancy) has the Cε methyl rotated about 90° and would allow a water molecule to approach the N5 atom; indeed, the occupancies of the Met 20 Cε methyl and the water are intricately linked to one another. Surprisingly, the Met 20 loop conformation in this structure is decidedly closed; this was previously thought to restrict solvent access to the substrate. However, it is more granular than that. The conformational disorder of a single residue, Met20 itself, seems to be responsible for solvent access, the Cε methyl group acting as a physical gate for water entry to N5 [44]. As compelling as this was, it did not provide direct evidence of a water molecule being the agent of substrate protonation. Unfortunately, the 2.0 Å nuclear density maps at pH 7.0 were unclear on this, as cancellation effects from the hydrogens on the Met20 side chain prevented us from being able to model it or a water molecule near to the N5 atom.

Thus, we attempted a “pH-trapping” experiment to catch a glimpse of the solvent species and possibly a path for substrate protonation. A similar crystal grown and HD-exchanged identical to the one used for the pH 7.0 neutron structure was vapor diffused for 1 week in a quartz capillary against a buffer at pH 4.5 and then was used to collect a 2.1 Å-resolution neutron data set [45]. Comparison of complementary room temperature X-ray structures revealed strikingly dampened dynamics of the DHFR structure at acidic pH, at both the global and residue level. Importantly, the Met 20 loop and Met 20 residue itself are both more ordered at pH 4.5. Furthermore, nuclear density could be observed on many ionizable residues suggesting protonation; indeed, all His and some Glu/Asp residues could be modeled as fully protonated (Figure 1). In the active site of the pH 4.5 neutron structure, a strongly positive and spherical m*F_o_*–D*F_c_* nuclear density peak could be observed adjacent to the Sδ atom of Met 20 on one side and a nearby D_2_O (known as the conserved DOD 47 molecule) on the other. Due to the strength and shape of the difference peak, we modeled a deuteron in the peak and its occupancy refined to ~0.8 with no resultant negative difference density. The deuteron sits between the Met 20 side chain and the DOD 47, at ~4 Å of distance between it and the substrate N5 atom (the target of protonation in the mechanism) (Figure 2F). Using the electron density maps, the Met20 side chain is more ordered; however, it could still be modeled in two alternate conformations. At pH 4.5, its side chain points away from the conserved water molecule (~40% occupancy) whereas the side chain points toward the water with about 60% occupancy. The minor conformation here would allow room for the deuteron, with the major conformer being in steric conflict with it. This rotational dynamic nature of Met20 is observed at both pH 4.5 and 7.0, suggesting mechanistic importance; this side chain may act as a solvent entry gate to the active site near the N5 atom to be protonated. We proposed that the deuteron observed likely is donated by the nearby conserved water molecule (D_2_O in the maps and known as DOD 47) and transiently exists as a D_3_O^+^ to perform this task. Complementary QM/MM calculations showed that a hydronium (H_3_O^+^) in the position occupied by DOD 47 rapidly and favorably transfers a proton to the substrate N5 [45]. The QM region was set up as the Michaelis complex, with DHF substrate and the nicotinamide moiety of NADPH bound in the active site; contrast with the “pseudo-Michaelis” complex bound in the crystal structures with a weak substrate (folate) and product cofactor (NADP^+^). We should note that the Met 20 loop is closed in these calculations and remains such during proton transfer to the N5 of DHF. In fact, the steric constraints of the active site in the QM region are such that only a proton could traverse from the H_3_O^+^ (DOD 47 in the NC structure) to the N5 atom of DHF. Thus, we propose that the DOD 47 is the catalytic water; however, its state as the agent of substrate protonation is as a transient hydronium (H_3_O^+^) ion. Solvent access to the active site is restricted by the closed Met20 loop; however, the conformational substates of the Met 20 side chain acts as a dynamic gate to specifically allow a water molecule (stabilized as a hydronium at acidic pH) near to the substrate. A proton dissociates from the hydronium and rapidly bonds with the N5 atom on DHF. This proton (deuteron) was only observed at acidic pH.

Two recent ultrahigh-resolution X-ray crystallographic studies of *ec*DHFR in its pseudo-Michaelis complex have provided support for the conformational heterogeneity of the Met 20 residue itself being important for solvent gating [46,47]. Both resolve two distinct side-chain conformers for Met 20 with nearly equal occupancy values, with one conformation allowing space for a partially occupied water molecule near N5 of the substrate and the other conformation sterically occluding solvent entry. By collecting data at two different X-ray beam energies to observe radiation-driven catalysis, Smith et al. (2024) showed that the occluded conformation population decreases while the water-permitting conformation increases, suggesting the importance Met 20-mediated water entry to the active site for catalysis [47]. Importantly, the authors point out that Met 20 is replaced by Leu 20 in *Bacillus subtilis* DHFR (*bs*DHFR), and in an ultrahigh-resolution structure of *bs*DHFR they do not observe the conformational heterogeneity of Met 20 in the *ec*DHFR structures. This suggests that *bs*DHFR uses a different mechanistic path for substrate protonation; it could still involve solvent as the agent of protonation but the specific entry point of a water to access the active site could be novel to *bs*DHFR [47].

## 3. Conclusions

Recent breakthroughs in structural biology include atomic-resolution electron cryomicroscopy, serial femtosecond and X-ray free electron laser crystallography, and AI-assisted structure modeling software that approaches empirical standards [48,49]. This has led to an unprecedented explosion in our knowledge of macromolecular structure. However, even with these powerful and exciting tools at our disposal, in every new structure, there is a hidden “layer” of information, the hydrogen atoms, that can only be consistently revealed by a neutron crystallography experiment. As these atoms compose half the atomic content in macromolecules and the majority of the atomic content in water, it is imperative for our complete understanding of both structure and function that these layers are revealed, that the invisible become visible.

The four vignettes presented here serve as examples of how neutrons can identify and distinguish various water species in enzyme structures. For xylose isomerase, hydronium (D_3_O^+^) molecules, deuteroxide (OD^−^) molecules, and deuterons (D^+^) could be modeled across several structures [18,19,20]. In PcyA bilin reductase, a D_3_O^+^ is found adjacent to the active site and triangulated between functionally important ionizable residues [29,30]. A D_3_O^+^ is proposed to stabilize a particular side-chain conformation and be a critical proton donor to Glu 177 in the glucoside hydrolase, Xyn II [33,34]. Finally, using a “pH-trapping” experiment, the NC structure of DHFR at acidic pH unveils a deuteron forming potential low barrier hydrogen bonds with a water molecule and a Met 20 side chain, near to the site of substrate protonation. It is hypothesized that this deuteron has dissociated from a transient D_3_O^+^ to be the proton donor necessary for one of the catalytic steps [44,45]. Although we should be encouraged by the emergence of these atom-level details and their ability to inform enzyme mechanisms, this is only a small sample size. Water molecules play invaluable roles in macromolecular function, especially in enzymes that perform acid-base hydrolysis; further NC structures from different enzyme families are needed.

## 4. A Memorial Reflection on Dr. Chris G. Dealwis (1964–2022)

“Seeing is believing”: we heard Chris say this quite a few times while working in his research lab, especially when he was justifying to one of us gathering more data or solving one more structure to add to a manuscript. Although we understood the gist of what he meant and knew the phrase was a well-intentioned cliché, it was rather strange for a scientific mentor to evoke the word *belief* about things that could be empirically determined. To be honest, we were focused on the wrong word. Chris was emphasizing the special *visual* power of structural biology to uncover molecular shape, conformation, interactions, and even perhaps a glimpse at function. Chris was trained as an X-ray crystallographer at Birkbeck College under the tutelage of Sir Tom Blundell and emerged into a rich age of discovery for structural biology, when recombinant DNA technology meant you could study just about any molecule and determine its structure. Why use a page of text to describe a car when I can just show you one? To be sure, a snapshot or a drawing of a car does not fully explain how it works; that’s where supportive text (and methods) can help. Chris understood this; in his career as an independent investigator at the University of Tennessee and then at Case Western Reserve University, he was always on the lookout for complementary techniques to support and/or supplement his X-ray structures. Though still a crystallographic method, neutrons proved an invaluable additional tool in our investigations into DHFR (see Section 2.4 above). Chris knew that we could answer important questions about drug binding and enzyme mechanisms if we could see the actual positions of the hydrogen atoms on the protein, ligands, and water molecules.

Across the years and several papers, “seeing is believing” became an unofficial catch phrase for the DHFR neutron project. Chris relished any moment he got beam time at an X-ray or neutron beam line, often staying up most of the night at the synchrotron with his students and postdocs helping to collect data. Even as Chris’s physical vision declined to where he was essentially blind, he maintained an active, well-funded research lab and continued to tenaciously lead important projects to unlock functional details found in molecular structures. It is indeed ironic, even tragic, that a person so dedicated to visualizing how the molecular machines inside our body work would suffer the loss of his eyesight. Even though we mainly knew Chris as a scientist and our PI, we also saw him as a loving husband to Martha, herself so supportive to us and especially to Chris after he lost his eyesight. Lastly, we also knew Chris as a man of strong Christian faith; he never pushed it on us, but he carried himself as someone with passion and with mercy for all of those around him. He is and will continue to be missed.

“Open your eyes and see what you can with them before they close forever”, Anthony Doerr (2014) All the Light We Cannot See, pp. 48–49.

## Figures and Tables

**Figure 1 biology-13-00850-f001:**
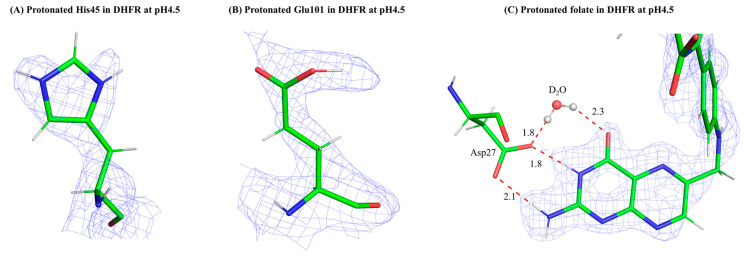
Neutrons are a powerful probe of macromolecular structure, revealing positions of hydrogens and thus protonation states of ionizable residues in proteins. 2*Fo–Fc* nuclear density (contoured at +1.0 σ, shown as blue mesh) of the protonated active site residues and folate substrate in DHFR at pH 4.5. (**A**) a protonated His residue; (**B**) a protonated Glu residue; (**C**) the protonated folate substrate bound in the active site. (All from PDB: 7D6G). Protein atoms are represented as sticks, with narrow diameter sticks for the hydrogens. Water molecules are represented as ball-and-stick. Atoms are colored as: green for carbon, red for oxygen, blue for nitrogen, and white for hydrogen. Bonding distances are shown in Å.

**Figure 2 biology-13-00850-f002:**
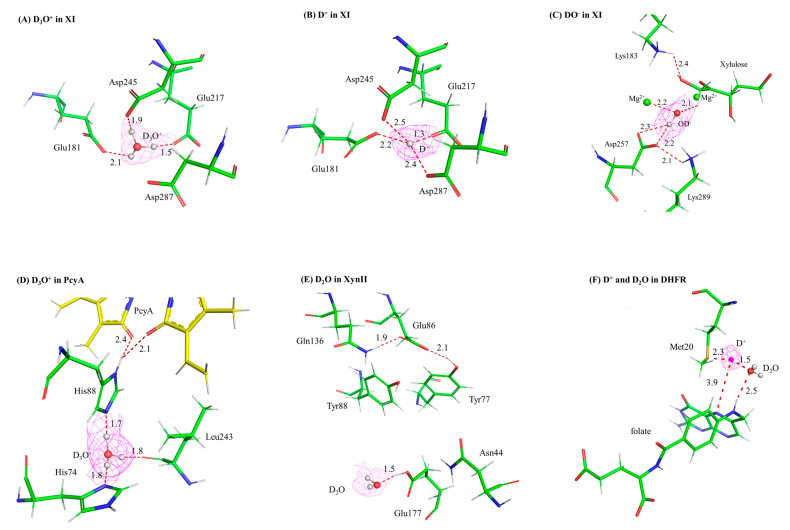
Neutrons are ideally suited for the identification of unique water species in macromolecular structure, especially catalytically important solvent molecules in enzymes. 2*Fo-Fc* nuclear density (contoured at +1.0 σ, shown as magenta mesh) of deuterated solvent (D_3_O^+^, D_2_O, DO^−^) and deuteron (D^+^) species in the XI, PcyA, XynII, and DHFR structures. (**A**) D_3_O^+^ in XI (PDB: 3KCJ), (**B**) D^+^ in XI (PDB: 3QZA), (**C**) DO^−^ in XI (PDB: 3CWH), (**D**) D_3_O^+^ in PcyA (PDB: 4QCD), (**E**) D_2_O in XynII (PDB: 4S2F), and (**F**) D^+^ in DHFR (PDB: 7D6G). Protein atoms are represented as sticks, with narrow diameter sticks for the hydrogens. Water molecules are represented as ball-and-stick. Atoms are colored as: green or yellow for carbon, red for oxygen, blue for nitrogen, white for hydrogen, and white or magenta for deuterium or deuterons. Bonding distances are shown in Å.

**Table 1 biology-13-00850-t001:** X-ray and neutron scattering profile of elements that compose proteins.

Element (Z)	X-Ray Scattering Length (fm), sin θ = 0	X-Ray Total Cross Section	Neutron Coherent Scattering Length (fm)	Neutron Incoherent Scattering Length (fm)	Neutron Total Cross Section
**^1^H (1)**	2.80	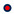	−3.74	25.27	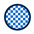
**^2^H/D (1)**	2.80	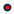	6.67	4.04	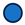
**^12^C (6)**	16.90	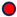	6.65	0.00	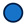
**^14^N (7)**	19.70	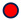	9.37	2.00	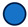
**^16^O (8)**	22.50	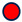	5.80	0.00	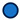
**^32^S (16)**	45.00	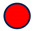	2.80	0.00	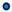

Relative cross sections for X-ray and neutron scattering are shown as circles, with red for X-ray cross sections and blue for neutron cross sections; this is for illustrative purposes only. Note that hydrogen (^1^H) has a large incoherent scattering cross section for neutrons, thus its total cross section is shown as a large sphere with a checkered pattern.

## Data Availability

No new data were created for this study.

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
