# Peer review of "“Seeing Is Believing”: How Neutron Crystallography Informs Enzyme Mechanisms by Visualizing Unique Water Species"

_biology, 2024, doi:10.3390/biology13110850_

Round 1

Reviewer 1 Report

Comments and Suggestions for Authors

This review explains the features of neutron crystallography, especially the visualization of unique water species. It will help readers understand the usefulness of neutron crystallography in elucidating enzyme mechanisms. However, the authors should revise the following points before publication.

Line 104-106: Ref. 7 (neutron structure of lysozyme) should be removed because this structure was actually determined at 1.8 Å resolution (not approach to 1 Å).

Section 2 (line 149-173): The authors should also briefly describe the observation of deuteroxide, e.g. neutron structure of copper-containing nitrite reductase determined at 1.5 Å resolution (Fukuda et al., (2020) Proc Natl Acad Sci U S A 117: 4071-4077 ).

Line 161: strength -> strengths

Author Response

Reviewer 1

Comments: "Line 104-106: Ref. 7 (neutron structure of lysozyme) should be removed because this structure was actually determined at 1.8 Å resolution (not approach to 1 Å)."

We thank the reviewer for pointing this out. While neutron data could be collected to 1.2 Å resolution, indeed the refinement was limited to 1.8 Å resolution. We have removed the mention of lysozyme and reference 7 from our reference list.

"Section 2 (line 149-173): The authors should also briefly describe the observation of deuteroxide, e.g. neutron structure of copper-containing nitrite reductase determined at 1.5 Å resolution (Fukuda et al., (2020) Proc Natl Acad Sci U S A 117: 4071-4077 )."

Thank you for pointing out the presence of a deuteroxide in nitrite reductase. We have added the following sentence in the XI section: "A deuteroxide has also been identified in a 1.5 Å resolution neutron structure a copper-containing nitrite reductase. Like XI, the deuteroxide forms a coordination bond with a metal (Cu2+ instead of Mg2+)." Please see lines 198-200. 

"Line 161: strength -> strengths"

Thank you. We have made the correction.

Reviewer 2 Report

Comments and Suggestions for Authors

This review article in memory of Dr. Chris G. Dealwis presents a very well structured and focused view on four interesting enzyme systems where neutron crystallography has provided key insight. The paper is very well written and easy to follow. The introduction is at the right level of detail and the descriptions of the four enzymes are succinct and to the point. I have however some minor comments and suggestions which are listed below:

1. While the authors' reasoning for avoiding the use of "deuteronium" is understandable, they could consider leaving the "deutero" names out altogether and just refer to e.g hydronium (as D3O+) as in ll. 159-161.

2. ll. 70-71 It is not quite correct to present perdeuteration as an alternative to H/D exchange, since they affect different atoms. Replacing "alternatively" on l. 70 with "Additionally" would for example make the statement correct. Also deuterons introduced by perdeuteration will not back-exchange. (ll. 73-75)

3. The authors consistently refer to Fo-Fc maps; I assume they should be mFo-DFc.

4. ll 235-237 When the authors refer to the putative D3O+ at the protein surface of rubredoxin, they could also comment on whether similar computational chemistry studies as for XI were performed on rubredoxin?

5. ll. 423-424 Hydronium is an ion, not molecule.

Author Response

"1. While the authors' reasoning for avoiding the use of "deuteronium" is understandable, they could consider leaving the "deutero" names out altogether and just refer to e.g hydronium (as D3O+) as in ll. 159-161."

We thank the reviewer for the suggestion. Yes, we struggled with this nomenclature diversity. As you suggest in the hydronium example, a possible remedy would be to just name it one way and refer to it consistently throughout the manuscript. An issue arises when describing what is being modeled in the nuclear density. For example, for hydronium, what is technically being modeled in the density is of course D3O+, so when one refers to the maps and the modeled ion, you are referring to deuterium atoms. Regardless, we admit this naming convention is a bit confusing. We would like to keep it as is if possible and just include the footnote 2 at the bottom of page 4. 

"2. ll. 70-71 It is not quite correct to present perdeuteration as an alternative to H/D exchange, since they affect different atoms. Replacing "alternatively" on l. 70 with "Additionally" would for example make the statement correct. Also deuterons introduced by perdeuteration will not back-exchange. (ll. 73-75)"

Yes, we agree, thank you for this point. We have substituted "Additionally" for "Alternatively" in the revised manuscript; please see line 70. We also include a parenthetical sentence on lines 76-77 addressing perdeuteration and back-exchange: "(Note: importantly, for perdeuterated samples, deuterons incorporated at non-exchangeable positions will not back-exchange.)"

"3. The authors consistently refer to Fo-Fc maps; I assume they should be mFo-DFc."

Yes, thank you, we did omit the map weighting factors of m (figure of merit) and D (sigma-A weighting). We have revised the manuscript to add these in. This is at multiple points in the manuscript, however, an example of a correction is line 110 on page 3. 

"4. ll 235-237 When the authors refer to the putative D3O+ at the protein surface of rubredoxin, they could also comment on whether similar computational chemistry studies as for XI were performed on rubredoxin?"

I haven't found a study that describes this. However, the D3O+ ions in rubredoxin are on the protein surface, not isolated in the protein interior, and thus are able to form extensive hydrogen bonding networks with surface water molecules. Thus, it may not be a valid comparison to the D3O+ observed in XI and (maybe) in PcyA.

"5. ll. 423-424 Hydronium is an ion, not molecule."

Thank you for pointing out this mistake, It has now been fixed. See lines 431-432.

Reviewer 3 Report

Comments and Suggestions for Authors

This review focuses on illustrating usefulness of neutron crystallography (NC). By utilizing the powerful ability of NC to visualize hydrogen (deuterium) atoms, water molecules that are incorporated into the mechanisms of protein functions can be visualized. Following the introductory explanation of how NC provides a unique and powerful tool to visualize water molecules, the authors illustrate, by describing four examples, how the visualized water molecules (hydronium ions, deuteroxide ions, deuterons) could help understand the mechanisms of enzyme actions. The description is concise and well organized.

It should, however, be noted that interpretation of the visualized scattering length density of water molecules should be done with caution because water molecules could adopt various shapes, including spherical, rod-like, and boomerang-like or triangular shapes, in the neutron Fourier maps, depending on the interactions with surrounding atoms (Chatake et al., Proteins, 50 (2003) 516). Thus, for example, when spherical (or rod-like) density is observed, careful consideration is needed to interpret if such density is really a deuteron (or a deuteroxide ion) or a part of a water molecule. It is therefore helpful how the observed density is concluded to be a deuteroxide ion (ll. 193-195, p.5), or a deuteron (ll. 212-216, p. 5, and ll. 400-402, p. 9) is explained more in detail.

The authors also describe the examples of observation of hydronium ions. However, the first example of PcyA (l, 238, p.6 - l. 302, p. 7) is rather confusing. Although the group conducted NC of PcyA insisted that a hydronium ion is observed (ll. 253-254, p. 6), the same group gave the contradict results by QM/MM calculations (ll. 266-269, p. 6). This cast a doubt on observation of hydronium ions. The second example of Xyn II (ll. 303-338, p. 7) is also confusing. The density in the map was proposed to be a hydronium ion (l. 312, p. 7), but described as “possibly” hydronium (l. 322, p. 7). It is unclear if a hydronium ion is really observed. It is helpful to clarify this point in these examples. 

Finally, the minor point: is the author name in the epigraph correct?

Author Response

Reviewer 3 Comments

  1. "It should, however, be noted that interpretation of the visualized scattering length density of water molecules should be done with caution because water molecules could adopt various shapes, including spherical, rod-like, and boomerang-like or triangular shapes, in the neutron Fourier maps, depending on the interactions with surrounding atoms (Chatake et al., Proteins, 50 (2003) 516). Thus, for example, when spherical (or rod-like) density is observed, careful consideration is needed to interpret if such density is really a deuteron (or a deuteroxide ion) or a part of a water molecule. It is therefore helpful how the observed density is concluded to be a deuteroxide ion (ll. 193-195, p.5), or a deuteron (ll. 212-216, p. 5, and ll. 400-402, p. 9) is explained more in detail."

Thank you for your attention to this. Indeed, interpretation and verification are very important parts of this type of analysis. We agree that we needed a bit more detail for this.

To address this, we have added the following sentences to the introduction in section 2 (now lines 172-180): "Several criteria are used to verify the identity of the water species in nuclear density maps, so as to distinguish between D2O, D3O+, OD-, and D+: 1) as mentioned above, the shape of the nuclear density, 2) the bonding and/or coordination contacts and binding distances in the binding site, 3) occupancy refinement of the atoms in the species, 4) examination of the mFo - DFc maps post-modeling and refinement to ensure difference density has been accounted for and no negative density has emerged, and 5) specifically for verification of a D+ would be the absence of a difference peak in electron density maps (as a deuteron would only very weakly scatter X-rays and are not observable in most electron density maps)."

Additionally, we did have some text already addressing this in section 2 already. For example, to confirm the D3O+ and the D+ in neutron structures of XI, please see section 2.1, lines 219-228.

2. "The authors also describe the examples of observation of hydronium ions. However, the first example of PcyA (l, 238, p.6 - l. 302, p. 7) is rather confusing. Although the group conducted NC of PcyA insisted that a hydronium ion is observed (ll. 253-254, p. 6), the same group gave the contradict results by QM/MM calculations (ll. 266-269, p. 6). This cast a doubt on observation of hydronium ions."

Yes, this wording was confusing. Thank you for pointing this out. The group responsible for the QM/MM calculations is independent from the groups who did the XI and the PcyA neutron structures. We have re-worded the sentence to better reflect that: "QM/MM calculations were performed to approximate the stability of the hydronium in PcyA [27]." See lines 278-280 in the revised manuscript.

3. "The second example of Xyn II (ll. 303-338, p. 7) is also confusing. The density in the map was proposed to be a hydronium ion (l. 312, p. 7), but described as “possibly” hydronium (l. 322, p. 7). It is unclear if a hydronium ion is really observed. It is helpful to clarify this point in these examples."

Thank you to the reviewer for bringing this to our attention. This was unintentionally misleading. The authors of the Xyn II neutron structure merely proposed a hydronium as important to the enzyme mechanism, however, they did not model and refine a hydronium in the nuclear density, at least they didn't publish any data to show this. We have added the following sentence to better reflect this: "It is important to note that the authors merely propose the existence of a transient D3O+ in this structure and do not model this water molecule as a hydronium; this is in contrast to the hydroniums identified and modeled in nuclear density for XI [21] and PcyA [30], as explained above." See lines 337-340 in the revised manuscript.

4. "Finally, the minor point: is the author name in the epigraph correct?"

Absolutely, thank you for pointing this out! This was a mistake and has now been corrected. See lines 40-41 in the revised manuscript.

Round 2

Reviewer 3 Report

Comments and Suggestions for Authors

The revised manuscript answers the concerns raised by this reviewer, and is now acceptable.